# Change of deep subduction seismicity after a large megathrust earthquake

Blandine Gardonio [1,2] ✉, David Marsan[3], Thomas Bodin [1], Anne Socquet [3], Stéphanie Durand [1], Mathilde Radiguet [3], Yanick Ricard [1] & Alexandre Schubnel[2]

Subduction zones are home to the world's largest and deepest earthquakes. Recently, large-scale interactions between shallow (0-60 km) and intermediate (80-150 km) seismicity have been evidenced during the interseismic period but also before and after megathrust earthquakes along with large-scale changes in surface motion. Large-scale deformation transients following major earthquakes have also been observed possibly due to a post-seismic change in slab pull or to a bending/unbending of the plates, which suggests the existence of interactions between the deep and shallow parts of the slab. In this study, we analyze the spatio-temporal variations of the declustered seismicity in Japan from 2000 to 2011/3/11 and from 2011/3/11 to 2013/3/11. We observe that the background rate of the intermediate to deep (150-450 km) seismicity underwent a deceleration of 55% south of the rupture zone and an acceleration of 30% north of it after the Tohoku-oki earthquake, consistent with the GPS surface displacements. This shows how a megathrust earthquake can affect the stress state of the slab over a 2500 km lateral range and a large depth range, demonstrating that earthquakes interact at a much greater scale than the surrounding rupture zone usually considered.

Subduction zones host earthquakes from 0 to 700 km depth that can be devastating. Past studies have introduced models to better understand the state of stress of the slab with the occurrence of intermediate and shallow seismicity[1–3]. Recent studies suggest that large subduction earthquakes can affect the plate stress regime by showing either synchronicity of the shallow and intermediate seismicity before large megathrust earthquakes[4–7], a slab-wide deformation transient concomitant with deep and shallow earthquake swarms before a large deep-focus earthquake[8] or a gravity field change[9,10]. The observation of large-scale surface displacement reversals, several months before megathrust earthquakes might also suggests the existence of deep precursors[11]. However, this is highly debated in the community since it might be due to processing artifacts. Furthermore, a postseismic increase in intermediate-depth seismicity down-dip of the ruptures of megathrust earthquakes in the upper plane of the double-seismic zone

has been evidenced in Japan[12] and in Chile[5], although this has recently been called into question by a new analysis[13]. One possibility to better understand these interactions is to analyze the seismic productivity as a proxy for the slab stress conditions after a megathrust earthquake[14,15]. In that respect, the 2011/3/11 $M_w$9.0 Tohoku-oki earthquake is an ideal case study, since it is the best-recorded megathrust earthquake of the instrumental period of seismology[16–21], and it occurred within one of the best-monitored regions of the world, Japan. Several studies conducted after the Tohoku-oki earthquake evidenced that large-scale processes active before the earthquake might have played a role in triggering the mainshock[4,9,22–28]. After the earthquake, there was a shift from predominant compression to tension in the forearc region off the coast[29–33]. The offshore GPS-Acoustic stations located above the main coseismic slip area have shown large landward post-seismic displacement due to an early viscoelastic relaxation of the oceanic mantle

[1]Univ Lyon 1, ENSL, CNRS, LGL-TPE, F-69622 Villeurbanne, France. [2]Laboratoire de Géologie, Département de Géosciences, École Normale Supérieure, CNRS UMR 8538, PSL Research University, Paris, France. [3]Univ. Grenoble Alpes, Univ. Savoie Mont Blanc, CNRS, IRD, University Gustave Eiffel, ISTerre, 38000 Grenoble, France. ✉e-mail: blandine.gardonio@univ-lyon1.fr

below the subducting plate[34,35]. Furthermore, segments adjacent to the Tohoku-oki earthquake experienced a landward increase of surface velocity that could either be due to the acceleration of the subducting plate[36,37] or to a viscoelastic response as suggested in other subduction zone[38]. The Tohoku-oki earthquake is thus characterized by both a possible large-scale preparatory phase and a large-scale post-seismic response.

Here, we investigate the intermediate and deep seismicity (from 150 to 450 km depth and even 680 km in the Izu-Bonin area) of the entire Pacific slab, over the 2500 km long Japanese island arc. The spatio-temporal evolution of the seismicity rate $\lambda$ (the number of earthquakes per unit of time) is analyzed using a probabilistic approach. Our study provides new observations on the consequences of a large mega-thrust earthquake on intermediate and deep seismicity, and hence on the change of the stress state of a large portion of the Pacific slab after the Tohoku-oki earthquake.

## Results and discussion
### Observed variations in rate of intermediate and deep seismicity
Our aim is to investigate the temporal and spatial evolution of the background seismicity, as a proxy for changes in loading rates. We work on 21 years of earthquake activity located in the subduction zone of the Pacific slab, recorded by the JMA (Japanese Meteorological Agency) catalogue from 2000 to 2021. To avoid a signal mostly affected by aftershocks and foreshocks sequences, we removed

dependent earthquakes following the method of Marsan et al.[39] (see Methods, Fig. S1–S3). Keeping only the independent background seismicity[40] evidences temporal and spatial changes in loading rates (Fig. S4). 144 reference points are defined over a depth-range of 80 km to 640 km (colored dots in Fig. 1). For each point, we select the 200 closest earthquakes with magnitude ≥3.5 which corresponds to the magnitude of completeness at these depths[41]. Note that there are fewer earthquakes in the Japan Sea, so the distance between the reference points and the related earthquakes reaches 600 km in this region, compared with 300 km for the other points (Fig. S5). For every set of earthquakes, we investigate the temporal evolution of the seismicity rate given by $\lambda = N/\Delta t$, where $N$ is the number of events occurring in the time period $\Delta t$ (see Methods). We compute the relative change of seismicity rate $\lambda$ at the time of the Tohoku-oki earthquake as $(\lambda_a - \lambda_b)/\lambda_b$, where $\lambda_a$ and $\lambda_b$ are the seismicity rates after and before (over 11 years) the Tohoku-oki earthquake. The effect of the Tohoku-oki earthquake on the seismicity rate at depth is far from homogeneous over the Japanese islands and we observe three main areas of opposite behaviors (Fig. 1a). Underneath the island of Hokkaido, the seismic rate increases, up to 30% between 150 and 450 km depth. This is confirmed by the time series of the earthquakes located at latitude >42°N (Fig. S6a, b). The maximum change is found at 200 km at 45°N latitude (Fig. 2a, b, red curve). We detect no significant effect of the Tohoku-oki earthquake on the deep seismicity under the Japan Sea (Fig. 1a). The time series of the earthquakes located between 35°N and

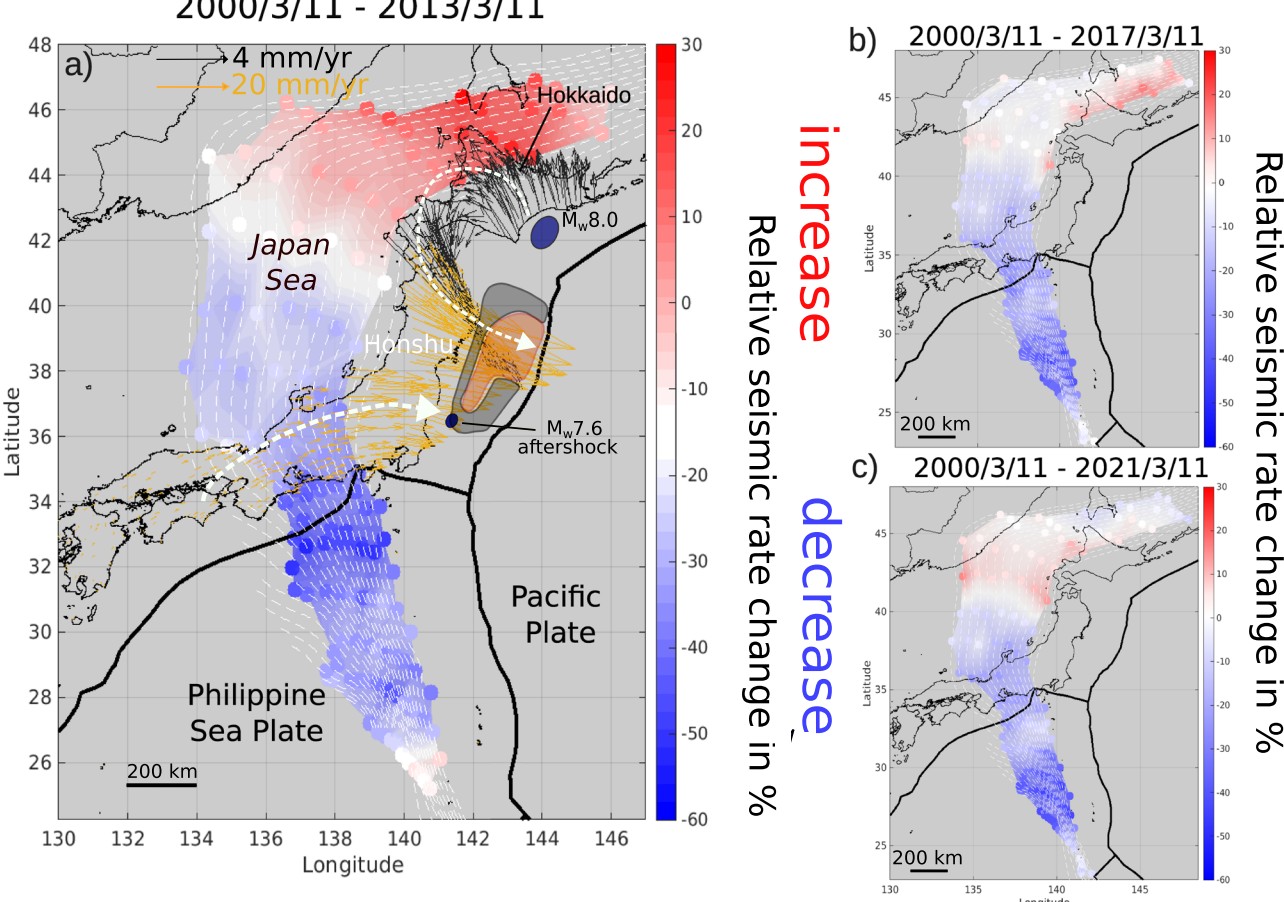

**Fig. 1 | Relative change in the rate of intermediate and deep seismicity of the Pacific plate taking the Tohoku-oki earthquake as a time of change.** We compare rate between 2000/03/11–2011/03/11 to the postseismic period. The data span **a** from 2000/03/11 (yy/mm/day) to 2013/03/11; **b** from 2000/03/11 to 2017/03/11 period, using the 260 closest events to sample the same area **c** from 2000/03/11 to 2021/03/11 period using the 320 closest events to sample the same area. Negative values (blue) indicate a decrease of seismicity rate, positive values (red) indicate an increase of seismicity rate. Every 20 km isodepth contours of the Pacific plate are shown from 80 to 400 km[34]. Black and yellow arrows give the GPS post-seismic displacement with different scales for stations above 42°N latitude and below. The orange and grey areas show the contours of the co and post-seismic slips[12].

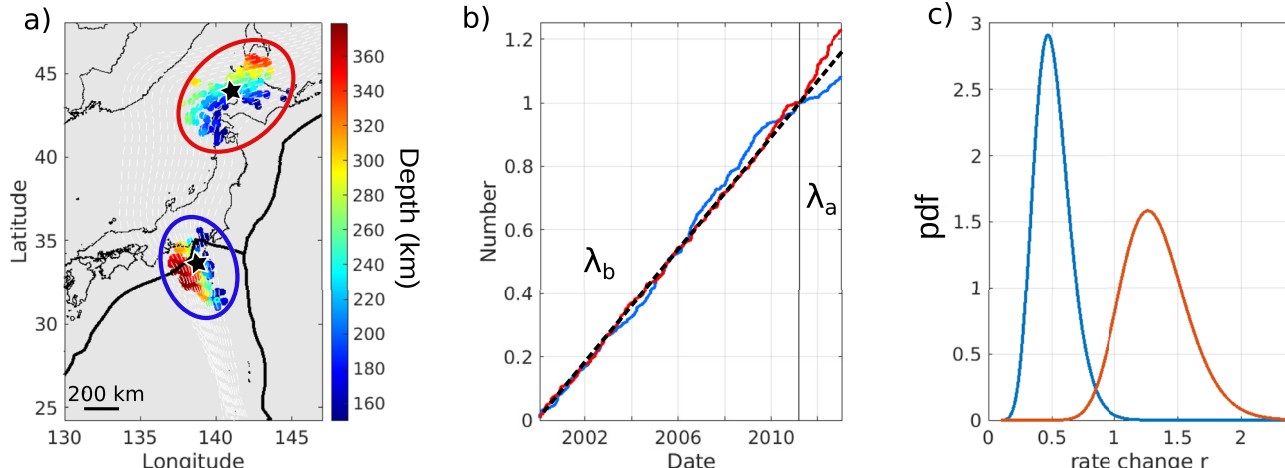

**Fig. 2 | Change of seismicity rate near latitudes 34°N and 44°N. a** Location of the earthquakes color-coded with depth. They correspond to the 200 closest earthquakes of the two black stars. **b** Cumulative number of earthquakes with time after declustering normalized at Tohoku-oki, for 34°N (blue) and 44°N latitude (red). Note the change of slope at the time of the Tohoku-oki earthquake, with $\lambda_b$ the slope before and $\lambda_a$, the slope after the megathrust earthquake. **c** probability density function (pdf) of the rate change (see Methods). A rate change r < 1, =1, >1, means decrease, no change and increase, respectively. The probability of change at Tohoku-oki is maximum with $r_{max}$ = 0.47 at 34°N (blue) and $r_{max}$ = 1.3 at 44°N latitude (red).

42°N shows no change of the seismic rate at the time of the Tohoku-oki earthquake (Fig. S6a, c). The southern part of the Japanese island arc, from 26° to 35°N latitude, shows a decrease of the seismicity rate by −55 to −10% with an average of −30% (Fig. 1a and S6a, d). The maximum change of −55% is found at 34°N latitude and 280 km depth (Fig. 2a, b, blue curve). By increasing the length of the analyzed period, the maximum change underneath Hokkaido and the negative change in the south attenuate over time until 2017 (Fig. 1b). The attenuation in the south continues until 2021, while there is a down-dip migration of the acceleration pattern in the north (Fig. 1c). The effect shows a relaxation with time, especially in the North but the relaxation is slower in the South (Fig. S7).

**Statistical significance of the variation in the seismicity rate**

In order to test the robustness of the observations, we perform various statistical tests on the onset of the seismicity rate change, focusing on the 2000/3/11–2013/3/11 period when we see the most significant changes due to the Tohoku-oki earthquake. We use different statistical approaches (see Methods), assuming that the rate of seismicity follows a non-homogeneous Poisson process. First, we analyze the probability of a change in seismicity rate, assuming that it occurs at the time of the Tohoku-oki earthquake. To that aim, we follow the formalism of Marsan and Wyss[42]. The probability density function (pdf) of the rate change r is shown in Fig. 2c. A rate change r less than or greater than 1 means a decrease or increase in seismicity rate, respectively. At 34°N latitude, the probability density for a rate change after the Tohoku-oki earthquake is maximum at $r_{max}$ = 0.45 meaning that it is most likely to have undergone a decrease of seismicity rate in that region at that time (Fig. 2c, blue curve). Conversely, an increase in the seismicity rate under the island of Hokkaido is confirmed with $r_{max}$ = 1.27 (Fig. 2c, red curve). It is possible to track this evolution by examining the pdf of the seismicity rate, $\lambda$ before and after the Tohoku-oki earthquake, separately (Fig. S8). We compute the standard deviation of r, $\sigma$, for a change at every grid point (Fig. S9). Plotting $r_m$, $r_{max}-\sigma$ (Fig. S9 left) or $r_{max} + \sigma$ (Fig. S9 right) leads to the same conclusions as we still see values close to 1 under Hokkaido for $r_{max}-\sigma$ and a decrease of the earthquake rate around 34°N latitude for $r_{max}+\sigma$. The high probability of having a change in seismicity rate at the time of the Tohoku-oki earthquake is thus confirmed whether we see an increase or a decrease of $\lambda$. We then focus on the probability of this change occurring at any other time. We again compute the pdf of r using the same dataset but now exploring

different times of change (Figs. S10 and S11). While $r_{max}$, i.e., the value that maximizes the pdf, is stable between 2000 and Tohoku-oki, there is a significant change of slope of the maximum at the time of the Tohoku-oki earthquake (black lines in Fig. S11a, b) implying that the regime changes at this precise moment, both at 34°N and 44°N latitude. More specifically, $r_{max}$ >1 between 2000 and the Tohoku-oki earthquake at 44°N latitude and $r_{max}$ < 1 at 34°N, showing the increase/decrease, respectively, of the seismicity rate at the time of the mega-thrust earthquake. Finally, we estimate the probability of having a change in the seismicity rate at any given time for the whole region by stacking the pdf of seismicity rate change calculated at each reference point (see Methods and Fig. 3). This probability shows a significant change at the time of the Tohoku-oki earthquake (Fig. 3a) indicating that a change in seismicity rate is most likely at the time of the Tohoku-oki earthquake, although its likelihood varies with latitude (Fig. 3b). Seeing no effect of the Tohoku-oki earthquake on the seismicity rate at the northern (above 45°N latitude) and southern tips (below 28°N latitude), confirms the spatial delimitation of our observations. Thus, the Tohoku-oki earthquake affected the stress state of the slab from the Izu-Bonin to the Hokkaido islands with a dichotomy between north (acceleration) and south (deceleration). We further clustered the 144 reference points according to their $r_{max}$ values at the time of Tohoku-oki earthquake applying the k-means clustering method (an iterative, data-partitioning algorithm that assigns *n* observations to exactly one of *k* clusters) considering 4 clusters (Fig. S12a). For each cluster, we display in Fig. S12b–e the mean of the pdf computed at each time step for all the points in the cluster. The first cluster is located underneath Hokkaido (Fig. S12b) and its $r_{max}$ value is maximum at the time of the mega-thrust earthquake (Fig. S12b). It corresponds to the area where the seismicity rate increases after the Tohoku-oki earthquake. Cluster 2 corresponds to locations where the seismicity rate is constant ($r_{max}$ = 1) (Fig. S12c). Cluster 3 gathers the southern points where there is a decrease of the seismicity rate after the megathrust earthquake. Its $r_{max}$ value is indeed minimal at the time of the Tohoku-oki earthquake. The same applies to cluster 4 with a smaller decrease, $r_{max}$ being close to 1. Note that all the grid points in the northern and southern tips belong to either cluster 2 or 4 which are those showing the mildest effect of the Tohoku-oki earthquake, hence outlining the spatial limit of the Tohoku-oki effect. This confirms the dichotomy between the northern and southern parts of the Pacific slab and the extent of the Tohoku-oki effect on the intermediate and deep seismicity.

After the Tohoku-oki earthquake, many small-magnitude earthquakes went undetected, especially at great depths, masked by the high productivity of larger, shallower, aftershocks. To address this issue, we analyze again the declustered JMA catalog taking a minimum magnitude of 4.5 and considering the 70 closest earthquakes to each reference points (Fig. S13). We still see the positive and negative changes underneath Hokkaido and at 34°N latitude, respectively. Note that the seismicity rate change in the Izu-Bonin at depth is still negative but less statistically significant. We finally carry out the same exercise (taking a minimum magnitude = 4.5 and the 70 closest events) using the USGS catalog and observe a strong increase underneath Hokkaido and a less visible decrease at 34°N latitude (Fig. S14).

## Discussion

We have shown that the changes of seismicity rate are statistically significant over the entire Japanese Pacific subduction zone between 150 and 450 km depth, and a spatial extent spanning 2500 km from 26° to 48° of latitude. The changes occur at the time of the megathrust earthquake. Interestingly, the maximum and the minimum rate changes are both at major bends of the Pacific subducting slab[43] (Fig. 1). Furthermore, the dip is larger in the Southern bend, around 34°N latitude, where the maximum negative variation was recorded, than underneath Hokkaido, where the maximum positive variation was recorded. The analysis of the Coulomb stress transfer on the area of the rupture zone of the Tohoku-oki earthquake, i.e., at a smaller depth than this study (0–150 km), highlights a cessation of the thrust earthquakes in the rupture zone that might last for centuries[14,15]. To get a first-order estimate of the elastic loading/unloading on the subduction interface, we compute the Coulomb stress change for two receiver points, where we observe the maximum acceleration and deceleration of the seismicity rate (orange and yellow stars in Fig. 4a). To do so, we use COULOMB3.3[44,45] that neglects the earth's sphericity. However, the use of a spherical earth model leads to differences in Coulomb stress change of only 10 to 25% on faults further away than those considered

here[46]. We impose a 50 m of slip on one dislocation located in the rupture zone of the Tohoku-oki earthquake. The receiver points are located according to the interface geometry[43] and we vary the rake angle from −180 and 180° for comparison with the rake values of faults in the surrounding area (Fig. 4b). We here make the hypothesis that a change in stress will affect the velocity of the slab and, by extenstion, the forces that apply inside it. A positive Coulomb stress change, i.e., would promote slip, is found for most dominant rake values at 44°N latitude, while a negative Coulomb stress change, an unloading, is found for most of the characteristics rakes at 34°N latitude coherent with an increase/decrease of the seismicity rate at these latitudes. We also analyze the Coulomb stress change using the focal mechanisms of the earthquakes closest to the two points located in Fig. 4 (Fig. S15 and Table S1 and S2). We also find mostly positive Coulomb stress change values for earthquakes underneath 44°N latitude and mostly negative stress changes for earthquakes underneath 34°N latitude, especially when considering the first nodal plane. We analyze 1439 GPS stations located all over Japan to compute the difference of slope in each time series, between a linear trend before the earthquake from 2008/01/01 to 2011/03/08 and a linear trend after the earthquake from 2011/03/13 to 2013/03/10 as we did with the declustered seismicity. We here do not seek to explain the details of the post-seismic relaxation but rather compute the change of direction of velocities. This confirms the enhanced landward motion that follows the Tohoku-oki earthquake observed in the Hokkaido region[36,37], which induces an anticlockwise rotation of the velocity change and an enhanced trenchward motion in Honshu. This landward GPS velocity change in Hokkaido is located right above the maximum positive change of the seismicity rate observed at 44°N latitude. Furthermore, we see a possible second, larger-scale, clockwise rotation of the displacement in the south, at 34°N latitude, where we observe the maximum negative change. Therefore, there is a clear spatial correlation between the surface displacements and the changes in the seismicity rate which places constraints on the location of the changes at great depth that GPS

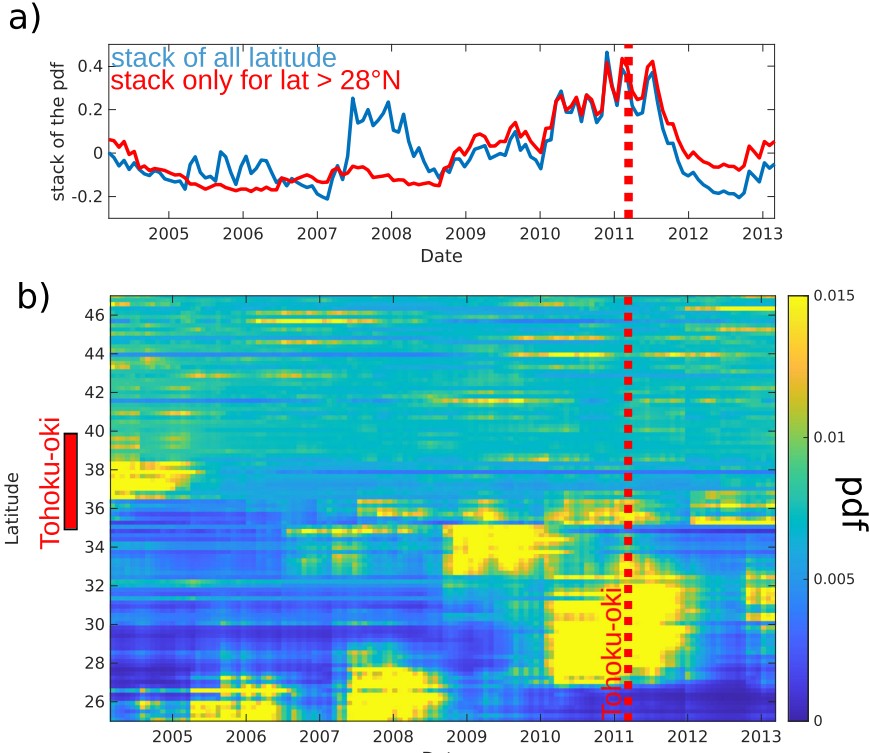

**Fig. 3 | Probability of having a seismicity rate change at any given time, stacked over all the reference points. a** Stack of all the data with or without points with latitude <28°. **b** Probability at each reference point plotted according to its latitude (see Methods).

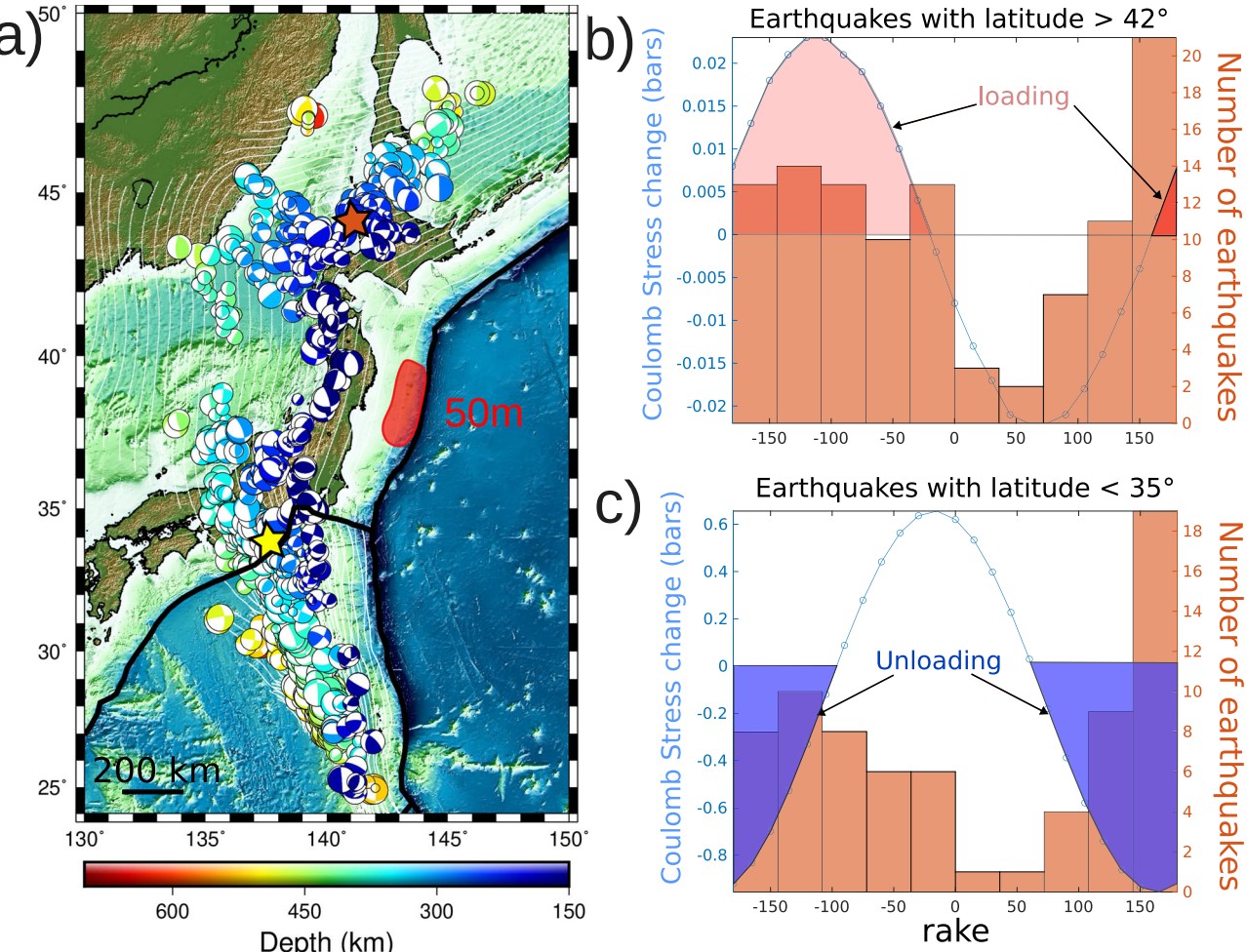

**Fig. 4 | Coulomb stress change computation. a** Focal mechanisms of intermediate and deep earthquakes that occur both before and after the Tohoku-oki earthquake (source from NIED https://www.fnet.bosai.go.jp/). The orange, 44°N latitude, and yellow, 34°N latitude, stars locate the points where we compute the Coulomb stress change due to coseismic elastic stress changes of **b, c. b** Blue: Coulomb stress change computed with COULOMB3 on a point located underneath Hokkaido, on the slab (lon = 140.8°, lat = 44.25°, depth = −204km, strike = 250°, dip = 26.7°, orange star in Fig. 4a) for different rake values, Orange: histogram of the rake values of all the earthquakes with latitude >42°, **c** Blue: same as **b** for a point underneath 34°N latitude on the slab (lon = 138.4°, lat = 34.70°, depth = −181km, strike = 161°, dip = 39°, yellow star in Fig. 4a); Orange: same as b) for earthquakes with latitude <35°. Here we assume a 0.4 friction coefficient.

alone cannot resolve. Note that a $M_w$7.6 aftershock of the Tohoku-oki earthquake, which occurred offshore the Boso peninsula, and its subsequent postseismic slip could be responsible for part of this velocity change[47]. Such rotation patterns might affect the interseismic velocity field of adjacent subduction segments[48] and different mechanical models could explain the enhanced landward motion seen in Hokkaido: an apparent increase in coupling in regions adjacent to the megathrust[36,37]; or a viscoelastic relaxation of both oceanic and continental mantle caused by coseismic and postseismic fault slip[34,35,49] that results in rotation patterns seen by GPS data[38].

Because the deep seismic response after the Tohoku-oki earthquake is both extremely fast and extensive, a rapid and large-scale stress transfer along the slab is required and could be facilitated by a low-viscosity channel, similar to the one modeled down to 135 km to explain the post-seismic deformation following the Maule earthquake[50]. Postseismic studies of the Tohoku-oki earthquake, suggest a low-viscosity shear zone[35,51,52]. These studies focus on the Honshu island only and the apparent change of coupling underneath Hokkaido remains to be modeled. As the variation in deep seismicity rate disappears slowly over time, this low viscosity is likely a transient feature, that can be modeled as a Burgers rheology combining Maxwell and Kelvin moduli commonly invoked

in viscoelastic post-seismic relaxation models, including after Tohoku-oki earthquake[29,51,52].

The Tohoku-oki earthquake has large-scale consequences on the very-deep (150–450 km and deeper in the Izu-Bonin area) seismicity of the Pacific plate. We observe significant changes in deep seismicity rate together with changes in surface displacement rates over a 2500 km lateral-range and up to 450 km depth, demonstrating that interactions between shallow and deeper parts of the slab exist at every stage of the seismic cycle. This questions our understanding and our vision of subduction zones that must be considered as a whole. In the future, taking greater depths into account in 3D rheological models and seismicity studies will bring more insights to the stress state of the slab and its continuity as a geodynamical object.

## Methods
### Declustering
We perform declustering of the earthquake dataset so to remove aftershocks that would otherwise contaminate the analyzed changes in seismicity rates. This is done based on the method already described in Marsan et al.[39], which uses an epidemic-type model of seismicity with a spatially heterogeneous background rate. We here provide the details of the method. Earthquake occurrences are modeled as a number of

earthquakes per unit time and unit area $\lambda(x, y, t)$, defined as the sum of two contributions:

$$\lambda(x, y, t) = \mu(x, y) + \nu(x, y, t)$$

in which $\nu$ accounts for triggering by previous earthquakes, and $\mu$ is the activity that would occur in the absence of any such interactions, i.e., the background rate, which we here assume to depend on position $(x, y)$ but not on time. Any earthquake with index $i$ occurring at $\{x_i, y_i, t_i\}$, with magnitude $m_i$, triggers aftershocks with rate:

$$\nu(x, y, t) = \sum_{i / t_i < t} \frac{K e^{\alpha m_i}}{(t + c - t_i)^p} \cdot \frac{\gamma - 1}{2\pi} \cdot \frac{L_i^{\gamma - 1}}{((x - x_i)^2 + (y - y_i)^2 + L_i^2)^{\frac{\gamma + 1}{2}}}$$

hence the product of the Omori-Utsu law with a power-law spatial density. Parameter values are fixed for all parameters based on either previous analyses or independent knowledge, except for $K$ which depends on completeness magnitude and the regional level of seismicity, and that, therefore cannot be fixed a priori (we describe below how the value of $K$ is determined). We set $\alpha$ to 2.0, which is a typical median value when considering previous studies, $p = 1$ which corresponds to the simple Omori's law, $c = 10^{-4}$ days (the exact $c$ value has no impact on our study as long as it is sufficiently less than several minutes), $\gamma = 2.0$ which lies in the interval already proposed by Felzer and Brodsky[53] and Marsan and Lengliné[54]. The aftershock zone radius is defined as $L_i = 0.2 \times 10^{0.5(m_i - 2.5)}$ (in km), which has the same $10^{0.5 \, m}$ scaling as in Utsu and Seki (1955)[55] and Van der Elst and Shaw[56]. This radius is 4 times larger than the rupture radius of a 30 bar stress drop earthquake[57].

The values of $\nu(x_i, y_i, t_i)/K$ are computed once and for all, making the rest of the computation fast with an Expectation-Maximization method. The probability that earthquake $i$ is a background earthquake is $\omega_i = \mu(x_i, y_i) / \lambda(x_i, y_i, t_i)$. We start by taking arbitrary (but non-zero) values of $\omega_i$, e.g., $\omega_i = 0.5$, and smooth these values to obtain the a priori background rate:

$$\mu(x, y) = \sum_i \omega_i e^{-\sqrt{(x - x_i)^2 + (x - x_i)^2}/l} \cdot \frac{1}{2\pi l^2}$$

where $\ell$ is a smoothing parameter taken equal to 50 km. Parameter $K$ is then computed as:

$$K = \frac{\sum_i 1 - \omega_1}{\sum_i F_i}$$

where $F_i = e^{\alpha m_i} (ln(t_e + c - t_i) - ln \, c)$, with $t_e$ the ending date of the studied period. This corresponds to the Maximum Likelihood Estimate of $K$ knowing $\omega_i$, for $p = 1$ as assumed here. Given $\mu(x, y)$ and $K$, the a posteriori probabilities $\omega_i$ can then be computed, and the procedure is thus iterated until all values eventually converge to their final estimates. The latter do not depend on the initial, arbitrary choice of $\omega_i$.

To demonstrate how the declustering affects the data, we show in Figure S1 the earthquake catalog for all magnitude ≥3.5 earthquakes, no condition on depth, before and after declustering. Figures S2 and S3 show the declustered earthquakes deeper than 150 km with a magnitude ≥3.5. The declustering method yields probabilities of being a background earthquake for each event as the final product. We here randomly draw background earthquakes based on these probabilities —for the purpose of simplifying the visual aspect of the plots. Figures S1–S3 demonstrate that visually evident aftershock clusters are indeed removed efficiently, which is the case for all aftershock sequences except the remarkable case of the 2011 Tohoku-oki main-shock itself (Figure S1). Here the aftershock sequence is only

attenuated but is still very visible in the declustered catalog. This is mainly due to our assumption that aftershock triggering is controlled by the epicentral distance between the mainshock and its aftershocks; this assumption is clearly not valid for an extended source like a M9 mainshockwhen performing an analysis at the regional scale as we do, hence the poor behavior of the method for this particular mainshock.

Figure S2 shows the selected deep earthquake dataset analyzed in this study, before and after declustering. This subset is not processed separately from the rest of the catalog: we decluster the catalog as a whole and afterward extract the selected (deep) subset out of the overall declustered catalog. Note that the aftershock sequence of the Tohoku-oki earthquake is not present at these depths. While 57% of all earthquakes are found to be aftershocks, this reduces to 38% when considering only the deep selected earthquakes. This drop is expected as intermediate and deep activity is less prone to large aftershock triggering[58,59]. Figure S3 offers a complementary view to Figure S2 by plotting the inter-event time vs date before and after declustering, showing that the aftershock sequences have efficiently been removed.

### Statistical analysis of the earthquake catalog

In this study, we make the assumption that the period of interest is divided into 2 periods with durations $[\Delta t_b, \Delta t_a]$, representing the period of time before and after the time of change, respectively, and characterized by constant seismicity rates $[\lambda_b, \lambda_a]$. In each of these periods, the rate of seismicity is assumed to follow a Poisson process, i.e., the probability of observing $N$ events in a period given the rate $\lambda$ is

$$p(N \vee \lambda) = \frac{(\lambda \Delta t)^N \exp(-\lambda \Delta t)}{N!}$$

**First statistical analysis.** we impose the time of change between the two periods at the Tohoku-oki earthquake occurrence and given observed $N_a$ and $N_b$, and given $\Delta t_a$ and $\Delta t_b$, we look for the values of $[\lambda_b, \lambda_a]$ that maximize $p(N_b | \lambda_b)$ and $p(N_a | \lambda_a)$, which yields: $\lambda_b = N_b / \Delta t_b$ and $\lambda_a = N_a / \Delta t_a$.

**Second statistical analysis.** for a given time of change between the two periods, the probability of observing a rate of $\lambda_b = N_b / \Delta t_b$ is:

$$p(\lambda_b) = \frac{\Delta t_b (\lambda_b \Delta t_b)^{N_b} \exp(-\lambda_b \Delta t_b)}{N_b!}$$

Following Marsan and Wyss[60], we compute the probability that the rate is increased by more than a given ratio $r$:

$$P\left(\frac{\lambda_a}{\lambda_b} > r\right) = \int_0^\infty d\lambda_b p(\lambda_b) \int_{r\lambda_b}^\infty d\lambda_a p(\lambda_a)$$

The pdf of $r$ is then $p(r) = -\frac{dP}{dr}$. Together with equation (1) this leads to:

$$p(r) = \frac{\Delta t_a^{1 + N_a} \Delta t_b^{1 + N_b}}{N_a! N_b!} r^{N_a} \int_0^\infty d\lambda \, \lambda^{1 + N_a + N_b} e^{-\lambda(\Delta t_b + r \Delta t_a)}.$$

Under the integral, we recognize the Gamma function $\Gamma(N_a + N_b + 2)$ which is the continuous prolongation of the factorial (i.e., $n! = \Gamma(n + 1)$), therefore

$$p(r) = \frac{(N_a + N_b + 1)!}{N_a! N_b!} \Delta t_a^{1 + N_a} \Delta t_b^{1 + N_b} \frac{r^{N_a}}{(\Delta t_b + r \Delta t_a)^{2 + N_a + N_b}}$$

**Third statistical analysis.** We want to compute the probability for the position of the change point to be at a given time $t_0$. If $N$ events occur at times $d = \{t_1, t_2, ..., t_N\}$ and the rate $\lambda(t)$ is variable with time, following

[Rasmusse][61], the probability to observe those events at these times is

$$p(d \vee \lambda(t)) = \prod_{i=1}^{N} \lambda(t_i) exp\left(-\int \lambda(t)dt\right)$$

where the integral is computed from the start to the end of the time interval under consideration. Our model $\lambda(t) = \lambda_b$ if $t < t_o$ and $\lambda(t) = \lambda_a$ if $t > t_o$ with $t_o$ the change point. The probability of having a change in the earthquake rate at $t_0$ given the observations, is

$$p(t_o \vee d) = \int \int p(t_o, \lambda_b, \lambda_a \vee d)d\lambda_b d\lambda_a$$

Using Bayes' Theorem, this becomes:

$$p(t_o \vee d) = \int \int p(t_o, \lambda_b, \lambda_a)p(d \vee \lambda_b, \lambda_a, t_o)d\lambda_b d\lambda_a$$
$$= p(t_o)p(\lambda_b)p(\lambda_a) \int \int p(d \vee \lambda_b \lambda_a, t_o)d\lambda_b d\lambda_a$$

where $p(t_o)p(\lambda_b)p(\lambda_a)$ is called $k$ in the following hence

$$p(t_o \vee d) = k \int \int \lambda_b^{N_b} \lambda_a^{N_a} exp(-\lambda_b \Delta t_b - \lambda_a \Delta t_a)d\lambda_b d\lambda_a$$

We now perform a change of variable where $u = \lambda \Delta t p(t_o \vee d) = k \frac{1}{\Delta t_b^{N_b+1}} \int u^{N_b} exp(-u)du. \frac{1}{\Delta t_a^{N_a+1}} \int u^{N_a} exp(-u)du$

Under the two integrals, we recognize again the Gamma function.

Using Stirling's approximation: $N_b! = \sqrt{2\pi N_b}\left(\frac{N_b}{e}\right)^{N_b}$

we get

$$p(t_o \vee d) = k \frac{1}{\Delta t_b^{N_b+1} \Delta t_a^{N_a+1}} \sqrt{2\pi N_b}\left(\frac{N_b}{e}\right)^{N_b} \sqrt{2\pi N_a}\left(\frac{N_a}{e}\right)^{N_a}.$$

Since $\left(\frac{1}{e}\right)^{N_a+N_b}$ is a constant, it does not depend on $t_o$, and lumping together all the constants in a new k factor, we finally obtain

$$p(t_o \vee d) = \frac{k\sqrt{N_b N_a} N_b^{N_b} N_a^{N_a}}{\Delta t_b^{N_b+1} \Delta t_a^{N_a+1}}$$

used to compute Fig. 3.

## GPS data processing

The data of the 1439 GEONET stations have been processed in double difference using GAMIT/GLOBK software[62]. For each day, the GEONET data were split into sub-networks of about 40 sites chosen to minimize the baseline between stations and improve the resolution of phase ambiguity. Sub-networks share 2 common sites with nearby other sub-networks to "tie" the solutions together, following a similar approach as that presented in the framework of the PBO project[63]. A reference tie network containing at least 1 station from every other sub-network is also constructed to provide additional stability to the entire network combination. For each sub-network, we reduce 24-hr measurement sessions to daily estimates of station position, choosing the ionosphere-free combination and fixing the ambiguities to integer values. We use IGS final products for the satellite orbits, satellites clocks and Earth orientation parameters (https://www.igs.org/products). Following Herring et al.[63], the orbit parameters are fixed to the IGS values. Ocean loading corrections are applied at each stations, using FES2004 (Finite Element Solution)[64] ocean tidal loading. The Vienna Mapping Function (VMF1)[65] is used to map the tropospheric delay in zenithal direction. The zenith delay is estimated every 2-hours and 1 gradient parameter is estimated per day. We apply atmospheric tidal and non-tidal loading correction, following Tregoning and van Dam[66] recommendation. The different sub-networks are then combined together into a single daily solution in a regional stabilization approach using a Kalman filter with GLOBK software to obtain loosely constrained daily solutions. Then daily solutions are combined into a multiyear solution to derive the time series, and to express the solutions in the ITRF2014[67] with a 7-parameter transformation using regional IGS sites. The daily GNSS position time series[68] are available at https://doi.osug.fr/staging/GNSS_products/GNSS.products.Japan.html. These GNSS position time series are then corrected from any jumps associated with documented material changes. The change of velocity before and after the Tohoku-oki earthquake is simply calculated as the difference of slope in each time series, between a linear trend before the earthquake from 2008/01/01 to 2011/03/08 and a linear trend after the earthquake from 2011/03/13 to 2013/03/10.

## Data availability

All data used in this study (catalog and focal mechanisms) are available from the Hi-net website https://hinetwww11.bosai.go.jp/.

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

## Acknowledgements

Earthquake data were kindly provided by the Japan Meteorological Agency in cooperation with the Ministry of Education, Culture, Sports, Science and Technology. We acknowledge the help of the National Research Institute for Earth Science and Disaster Prevention, Tsukuba, for making available their waveform data. The authors would like to thank Cécile Lasserre, Pascal Bernard, Claudio Satriano, Hélène Lyon-Caen and Marianne Métois for insightful discussions. We acknowledge support from the European Research Council (grant number 681346-REALISM and 865963 DEEP-trigger).

## Author contributions

B.G. conceived the study. B.G., D.M., T.B., S.D., Y.R. developed the statistical methods, A.S. and M.R. analyzed the GPS data. A.S. cross-examined the observations and results. All authors participated in the writing of the manuscript.

## Competing interests

The authors declare no competing interests.
