## [Peer Review File · Nature Communications]

Change of deep subduction seismicity after a large megathrust earthquakeEditorial Note: Parts of this Peer Review File have been redacted as indicated to remove third-party material where no permission to publish could be obtained.

REVIEWER COMMENTS

Reviewer #1 (Remarks to the Author):

Gardonio et al. Show some very interesting rate changes in deep seismicity that hints at the role of a larger part of the plate interface than just the rupture zone during the seismic cycle. This work has important implications in terms of how we monitor subduction zones in particular suspected mature segments because it suggests that we need to consider the information measurements not only at the mature segment but also at adjacent segments of the subduction zone.

I think that all the argumentation until the Coulomb analysis is sound. I have major reservations about the Coulomb analysis because of the incorrect flat Earth assumption and also because of the oversimplicity of the stress transfer model that does not factor in the role of the other parts of the plate interface during the coseismic phase.

I recommend major revisions. I think major revision could be achieved by either stopping the paper after the discussion of seismic rate changes, or re-doing the stress change analysis using a model with Earth curvature and experimenting with coseismic kinematics on portions of the plate interface that are not in the mainshock rupture zone.

General comments:

Which seismic stations were used to build the seismic catalog?

The authors do a detailed job of explaining the declustering. However, I was wondering what the influence of changing seismometer distribution has been on the catalog on which the declustering was performed. This becomes less important with increasing the chosen magnitude of completeness, but still I think it should be explained in the manuscript.

Coulomb stress transfer analysis

The Coulomb 3.3 calculations seem to be done with a flat earth assumption but the study area is too large for flat Earth to be used.

Page 17 of the Coulomb 3.3 software, (<https://pubs.usgs.gov/of/2011/1060/of2011-1060.pdf>) it is stated: "Coulomb 3.3 calculates in the (x, y, z) Cartesian coordinate system"

Whether this is Earth Centred Earth Fixed (ECEF) or approximating a flat earth is not clear, although from figures of the software manual, I assume this is a flat Earth model.

The only way that I see around this flat Earth problem, is to make a mesh in ECEF coordinates, impose a slip, then extract the tractions on the plate interface at the analysed regions. This would be a lot of extra

work, and I am NOT suggesting that the authors should do this. Rather, I suggest that this is the only way (that I am aware of) to do such a stress change analysis when ignoring Earth curvature becomes inappropriate, given current available modeling software in our community.

Role of the plate interface during the coseismic

Another part of the Coulomb stress transfer analysis is the assumption that the kinematics of the coseismic are confined to the rupture area. The stress changes shown in figure 4b are tiny. Compare this to the stress changes in the Melnick et al. 2017 paper: They use a thermomechanical model and get much larger stress changes on mainshock-adjacent segments than those that can be achieved with basic Coulomb failure stress transfer modeling. Implicit in the model used by Melnick et al. is that the whole plate interface, not just the mainshock zone is kinematically active during the coseismic. See also Mavrommatis et al. 2014 in which there is coeval slip and backslip increases. If this can be occurring interseismically, why not also coseismically and postseismically?

Specific comments:

Abstract: "We observe rate of intermediate to deep..." This sentence is missing some time information in relation to the coseismic

Page 20: Linear trend pre Tohoku-oki and linear trend after. But the trend is not in the data? It is the trend that is in the 1st order polynomial model. Therefore why not do the cumulative displacement for an interseismic period and cumulative displacement for a postseismic period? Then we don't have to worry about the issue of mis-modelling the curvature of the time series in the postseismic.

Figure 3: Why are the probabilities high for rate changes at higher latitudes (where we see change in rate beneath Hokkaido)?

Figure 4: from subplots b) and c) there seems to be little correlation between delta coulomb failure stress and number of earthquakes. Also, in the plot and caption there is no mention that this analysis is for events after Tohoku-oki. While this might be obvious to some, it should be stated. Also, shouldn't this plot and the discussion of this plot on page 8 rather plot rate changes (postseismic compared to interseismic) in the histogram instead of cumulative number of events?

Reviewer #2 (Remarks to the Author):

This paper puts forward the very interesting possibility that intermediate-depth and deep-focus earthquakes in subduction slabs far away from the M9 Tohoku-oki earthquake experienced substantial and enduring increases and decreases in seismicity rate out to great distances. This is certainly a topic well worth exploring. However, I have to admit that I ended up finding myself somewhat frustrated, as I found it difficult to follow some of the descriptions in the paper (as detailed below). In several places, not enough explanations are provided of what seem to be important aspects of the analysis (e.g., removal of data affected by great earthquakes is only mentioned in a suppl. Figure caption). As currently presented, I came away not being convinced yet that the proposed rate changes are significant. I am listing a number of concerns below, and various additional comments, questions and edit suggestions are in the annotated manuscript I am returning.

In the hope of a clarified version being more convincing, I recommend major revisions of the current manuscript, allowing for reworking and strengthening of the analysis. I would be glad to review a revised version and the authors could also contact me directly if I may have misunderstood or missed something important.

Itemized comments (unfortunately there were no line numbers in the manuscript file, but I hope the annotated manuscript from which these comments were derived will allow for clarifying these):

p. 2, Abstract:

Not sure if this point about (still tentative) precursor suggestions is the best way to start off this paper. Why are you leaving out postseismic relaxation of coseismic stress changes in the mantle, which is the best documented process producing post-earthquake deformation transients?
Use "background rate" here and/or elaborate to note that this is looking at "declustered seismicity".

p. 3 - top: As for the abstract, I am not sure that starting out talking about proposed precursory seismicity, deformation and gravity-field changes before large events is a good way to start out this paper. As noted in the next sentence, the paper goes on to explore seismicity as a proxy of stress conditions after the Tohoku-oki earthquake, not before. Nonetheless, the cumulative event number plots presented in this paper don't seem to suggest notable rate changes in slab seismicity that could be related to the proposed months-long or multi-decadal precursors, right?
The studies noted here do not describe a (single) "preseismic phase", but range from investigations of decadal variations in plate interface coupling to (still debated) months-long slab deformation transients inferred from GRACE and GNSS time series.

p. 4: Would it be possible to assess how sensitive the results are to the choice of declustering approach (using method of Marsan et al. , 2017) by comparing the results with those based on other approaches/algorithms (e.g., Zhuang et al. 2002, Zaliapin and Ben-Zion, 2015).
Are there depth-dependent and lateral changes in M_c that could affect the results?
"We detect no significant effect of the Tohoku-oki earthquake on the deep seismicity" Can you please specify what you mean by "deep seismicity" in this sentence; i.e., what depth range (> 150 km?).

p. 8: But isn't the deep seismicity not on the plate interface (or slab top, but in seismic planes of the double seismic zone within the plate with contrasting mechanisms and nodal planes very different from the slab top (e.g., Kita et al., 2010)? That is, it is not a matter of rake variations only. There is no "fault" at the top of the slab that these events relate to.

First time GPS analysis is mentioned? Shouldn't there be an introduction of the data?

When considering the postseismic GPS motions, it is not clear if the authors look for processes above and beyond the postseismic relaxation that model studies put forward to explain most of the deformation signal. Are co- and postseismic stress changes relevant?

"viscoelastic relaxation of both oceanic and continental plates" : The viscoelastic relaxation occurs mostly in the mantle asthenosphere in the mantle wedge and below the oceanic plate (e.g., Freed et al., Hu et al., 2016), not in them as implied in this sentence. Good to clarify this.

"and could be facilitated by a low-viscosity channel," Is this envisioned low-viscosity channel on top of the slab? The coseismic stress changes that could drive such shear under Hokkaido must be quite small. It would be good to assess this scenario in a (simple) model. Some postseismic studies (e.g., Hu et al., 2016) did include a low-viscosity shear zone to represent such deep localized shearing. Unfortunately, this discussion of a low-viscosity channel and possible rheologies involved comes across as highly speculative without actual modeling to test this idea.

Page 15:

I am not a statistical seismologist, but this method seems sound. Based on my own experience, it is always useful to consider how the choice of declustering method affects the results. I think it would be worth running some comparisons.

Can you briefly elaborate on what this comparison implies?

P. 16: M 2.5 is well below M_c . How does using an incomplete catalog come into play? Again, it would be good to evaluate spatio-temporal variations of M_c to avoid biases in the analysis.

"except the remarkable case of the 2011 Tohoku-oki mainshock itself": Wouldn't this "imperfect" declustering of the M9 aftershocks potentially affect the results of this study, possibly introducing a bias?

"Figure S4 shows the selected deep earthquake dataset analyzed in this study, before and after declustering". Do you mean Figure S3? What is the "subset" mentioned here. In caption of Fig. S5 you "Note that we remove the area close to the Tohoku-oki and the Mw8.0 2003 Tokachi-oki rupture", but I think this is not explained otherwise in text. Is the gap in events at the 37-40 latitude range in Fig. S3 due to this? What does Fig. S3 look like applied to the original catalog of events?

What is Figure E? Do you mean Fig. S4?

p. 20: "The change of velocity before and after the Tohoku-oki earthquake are simply calculated as the difference of slope in each time series, between a linear trend before the earthquake from 2008/01/01

to 2011/03/08 and a linear trend after the earthquake from 2011/03/13 to 2013/03/10."

A linear trend and average velocity are not a very good representation of the rapidly decaying postseismic deformation in the first 2 years.

Figure 2: The two example time series show similarly large changes in rate at times other than the M9, including a rate increase in 2006, a much larger decrease starting in ~2008 for 34N and sudden decrease in early 2010 for 44N. Do you consider these to be significant?

Figure 4: Figures and their captions should be as self-explanatory as possible. Can you please elaborate in the caption on what this stress change is due to (coseismic elastic stress changes).

The assumption of receiver faults being on the slab surface and looking at varying rake angles alone is not warranted for intraplate deep-focus earthquakes.

Fig. S1: Catalog is incomplete below $M_c = 3.5$. Could spatio-temporally variable M_c affect these plots and analysis.

Good to include figures like Fig. S1 and S2 spanning full time period. Are events in these two figures same as Fig. 4, but prior to removing events near 2003 and 2011 mainshocks? Why not make more panels and show all the data available in this fashion.

These are also deep earthquakes? Point to evidence of M9 aftershocks not having been removed. What is the "subsetting" procedure to produce Fig. S3? Why not make a version of Fig. S3 with all data prior to this procedure, so we can better see the M9 aftershock effect?

"Fig. S3. Deep earthquakes selected in this study," : Please make clear if earthquakes in Figs. S1 and S2 not deep earthquakes. It seems that this would be the most important figure in the paper, given that this shows all the data this study is based on, right?

Why are there almost no earthquakes between ~37-40N, i.e., the latitude range of the M9 mainshock. Only after I get to the caption of Fig. S5 do I see mention of this representing a subset.

Fig. S5: Is this plotting the "subset" of data shown in Fig. S3? If so, why do we still see strong rate increase W of the M9 rupture zone?

It is rather disconcerting that the rate increase in the N is invisible prior to declustering. What does this figure look like without the removal of events near 2003 and 2011 mainshocks? Declustering should make a lumpy quake distribution smoother, by removing aftershocks concentrated near their mainshocks. That's not what we see here, and the entire pattern changes, with red turning blue and blue turning red, while the overall smoothness is about the same or a little improved. I realize this is likely related to the sampling scheme around the reference points.

This is not described in the text. Does this procedure explain the gap in events in Fig, S3? What are the exact areas that got removed. Why also remove the Tokachi earthquake area, given how early this event was?

Figure S6: Can you add a 300-km circle in lower right to provide a sense of scale for these sampling area.

What is the actual number of events within each of these latitude windows. What do cumulative number plots look like if rather than stacking the greatly overlapping samples around the reference points, you simply provide the count for each latitude interval? That seems like a more straightforward way of displaying the results.

Just looking at deviation from straight line (drawn in), rates seem to start to increase in ~2009, with some variability.

Fig. S12/S13: Is this analysis of $M > 4.5$ events also on declustered events and after removing events near 2003 and 2011 mainshocks? As I suggested there, can you repeat this analysis by simply plotting cumulative number of all events in the three latitude ranges shown in Fig. S6.

Fig. S12 & S13: Even though they are supposed to both show $M > 4.5$ events and include 70 events, there are about twice as many events in the NEIC catalog plots than the JMA catalog shown here. Why is that?

Fig. S13: Both areas seem to show a rather pronounced rate decrease in the (~5) years before Tohoku-oki.

Roland Burgmann

First of all, the authors would like to thank the editor and reviewers who greatly helped improving this study. We were very pleased to see many questions and comments since the observations we present are seen for the first time and can be quite puzzling. We have endeavored to clarify the computation of the Coulomb stress change, the declustering method and the consequences on our results of choosing a given magnitude of completeness. Also, we improved the abstract and introduction by presenting how the deeper part of the subduction could interact with the shallower part during the interseismic and post-seismic periods. Our changes to the manuscript are highlighted in red. We answer the reviewers with more details below.

10

REVIEWER COMMENTS

Reviewer #1 (Remarks to the Author):

15

Gardonio et al. Show some very interesting rate changes in deep seismicity that hints at the role of a larger part of the plate interface than just the rupture zone during the seismic cycle. This work has important implications in terms of how we monitor subduction zones in particular suspected mature segments because it suggests that we need to consider the information measurements not only at the mature segment but also at adjacent segments of the subduction zone.

20

I think that all the argumentation until the Coulomb analysis is sound. I have major reservations about the Coulomb analysis because of the incorrect flat Earth assumption and also because of the over-simplicity of the stress transfer model that does not factor in the role of the other parts of the plate interface during the coseismic phase.

25

I recommend major revisions. I think major revision could be achieved by either stopping the paper after the discussion of seismic rate changes, or re-doing the stress change analysis using a model with Earth curvature and experimenting with coseismic kinematics on portions of the plate interface that are not in the mainshock rupture zone.

30

General comments:

35

Which seismic stations were used to build the seismic catalog? The authors do a detailed job of explaining the declustering. However, I was wondering what the influence of changing seismometer distribution has been on the catalog on which the declustering was performed. This becomes less important with increasing the chosen magnitude of completeness, but still I think it should be explained in the manuscript.

40

Thank you for raising this point. In this study, we use the seismic catalog provided by the Japanese Meteorological Agency which uses the entire Japanese network to issue its bulletin (https://www.data.jma.go.jp/svd/eqev/data/bulletin/index_e.html). Given that this is one of the best instrumented areas in the world, this catalog presents a very low magnitude of completeness (as low as 2.5 event at the depths > 100km that we considered, see Figure 1 extracted from Schorlemmer et al., 2018). The Japanese network is very stable since about 2000 (Schorlemmer et al. 2018, Figure

45

1) so the quality of the data is not a topic of concern to us. We added a sentence in that sense in the
50 manuscript l.74.

[REDACTED]

Figure 1: Detection completeness of the Japanese seismic network, computed for 1 January 2015 for four different depth levels: 0, 30, 100, and 500 km (from Schorlemmer et al., 2018).

55 *Coulomb stress transfer analysis*

The Coulomb 3.3 calculations seem to be done with a flat earth assumption but the study area is too large for flat Earth to be used.

60 *Page 17 of the Coulomb 3.3 software, (<https://pubs.usgs.gov/of/2011/1060/of2011-1060.pdf>) it is stated: “Coulomb 3.3 calculates in the (x, y, z) Cartesian coordinate system”*

Whether this is Earth Centred Earth Fixed (ECEF) or approximating a flat earth is not clear, although from figures of the software manual, I assume this is a flat Earth model.

65 *The only way that I see around this flat Earth problem, is to make a mesh in ECEF coordinates, impose a slip, then extract the tractions on the plate interface at the analysed regions. This would be a lot of extra work, and I am NOT suggesting that the authors should do this. Rather, I suggest that this is the only way (that I am aware of) to do such a stress change analysis when ignoring Earth curvature becomes inappropriate, given current available modeling software in our*

70 *community.*

This is indeed a very important point. As we mention in the manuscript, the aim of the Coulomb Stress change analysis here is not to provide accurate values on the changes observed on the faults but rather to discuss about the positive and negative stress change according to the most common
75 fault geometries present in the studied areas and comparing it to the increasing or decreasing seismic rate. As the reviewer says, taking into account the geometry of the slab is very complex and out of the scope of this study and we are not convinced that it would significantly change the results.

For example, to model the co-seismic displacement in China due to the Tohoku earthquake, Chen et al, 2020, have shown that the use of a dislocation theory for a spherical Earth gives results closer to the observed co-seismic displacement but the direction and magnitude obtained with a half-space dislocation model are already in good agreement with the observed data. When looking at the stress on specific target faults in China, these authors find that the difference between a flat Earth and a spherical Earth models amounts to typically 10% to 25%, but that the sign of the Coulomb stress on
85 those faults is not changed (Figure 2).

90

[REDACTED]

95

Figure 2: Coulomb stress changes caused by the Tohoku-Oki Mw9.0 earthquake on the northern Tanlu fault zone at depths 20 and 25km by using the sphere earth dislocation theory (top) and the semi-infinite space dislocation theory (bottom)

100 We moreover note that these faults in China are further away from the M9 rupture than the two zones of interest in our study, ie, Hokkaido and Chiba, so that the error will be less. We now provide in the main text a short discussion on this issue, using the same arguments as above l. 168-170 (ie, mainly exploiting the Chen et al.'s results).

Another point is that it is very common to compute Coulomb Stress change after a large earthquake and we think that this computation will be expected by readers. Again, we do not pretend to provide accurate values but only the sign is important.

Role of the plate interface during the coseismic
Another part of the Coulomb stress transfer analysis is the assumption that the kinematics of the coseismic are confined to the rupture area. The stress changes shown in figure 4b are tiny. Compare this to the stress changes in the Melnick et al. 2017 paper: They use a thermomechanical model and get much larger stress changes on mainshock-adjacent segments than those that can be achieved with basic Coulomb failure stress transfer modeling. Implicit in the model used by Melnick et al. is that the whole plate interface, not just the mainshock zone is kinematically active during the coseismic. See also Mavrommatis et al. 2014 in which there is coeval slip and backslip increases. If this can be occurring interseismically, why not also coseismically and postseismically?

This is a very good question and the reviewer's remark to consider post-seismic load is pertinent. Melnick et al., 2017 have considered stress changes on areas that are much closer to the rupture zone than the areas analysed in this paper. We thus expect that the changes we compute would be smaller than that in Melnick et al., 2017.

Furthermore, the changes of seismic rate that we observe are sudden and last for years in a consistent way (i.e., the initial trend continues for several years). It thus seems that the coseismic perturbation is the triggering element. The post-seismic phase might also play a role in the long term but attributing the effect to coseismic, postseismic or both is out of the scope of this study which focuses on the effect of the megathrust earthquake on the deep seismicity. To better understand the potential role played by the coseismic and/or post-seismic slip, developing appropriate models would be a good follow up of this study.

Specific comments:

130 *Abstract: "We observe rate of intermediate to deep..." This sentence is missing some time information in relation to the coseismic*

We added the time information in l.23-24

135 *Page 20: Linear trend pre Tohoku-oki and linear trend after. But the trend is not in the data? It is the trend that is in the 1st order polynomial model. Therefore why not do the cumulative displacement for an interseismic period and cumulative displacement for a postseismic period? Then we don't have to worry about the issue of mis-modelling the curvature of the time series in the postseismic.*

145 In this study, we do not aim to model the postseismic phase but rather to measure the sign of the change of velocity of the GPS data, using the same approach as for the seismicity rate analysis, ie a difference of the linear approximations after vs before the Tohoku-oki earthquake.

Figure 3: Why are the probabilities high for rate changes at higher latitudes (where we see change in rate beneath Hokkaido)?

150 Thank you for this question and observation. It is true that the change of seismicity rate in the North is less important than in the South (+30 % and -50 %, respectively) and that it is more variable in the North. Actually, there is a long-term acceleration of the seismic rate in this area (see Figure S7 b). We see that the effect of the Tohoku earthquake is large enough when we remove that trend. That is why we observe a high probability for rate change at these latitudes. This is what we see in
 155 Figure S11 on the right with a r value being larger than 1 before the Tohoku-oki earthquake. This means that, for any given time between 2000 and the Tohoku-oki earthquake, the seismicity rate increases. The value of r is highest at the time of the megathrust earthquake showing that the seismicity rate increase is high (35% of increase). Then, r is still higher than 1 up to 2012 where we see a deceleration (see Fig. S10a).

160
 Figure 4: from subplots b) and c) there seems to be little correlation between delta coulomb failure stress and number of earthquakes. Also, in the plot and caption there is no mention that this analysis is the for events after Tohoku-oki. While this might be obvious to some, it should be stated. Also, shouldn't this plot and the discussion of this plot on page 8 rather plot rate changes (postseismic compared to interseismic) in the histogram instead of cumulative number of events?
 165

Actually, these are pre- and post-Tohoku earthquakes since the focal mechanisms have not changed much at these depths after the megathrust earthquake (see Figure 3). As we mention above, this Figure is here to show that the possible loading and unloading computed with the Coulomb stress change, even if not perfect, is coherent with the most common focal mechanisms found in the north
 170 and in the south.

Figure 3:
 Rake values for earthquakes used in Figure 4 of the manuscript. There are no significant changes of rake observed before and after the Tohoku-oki earthquake.

175

Reviewer #2 (Remarks to the Author):

180 *This paper puts forward the very interesting possibility that intermediate-depth and deep-focus earthquakes in subduction slabs far away from the M9 Tohoku-oki earthquake experienced substantial and enduring increases and decreases in seismicity rate out to great distances. This is certainly a topic well worth exploring. However, I have to admit that I ended up finding myself somewhat frustrated, as I found it difficult to follow some of the descriptions in the paper (as detailed below). In several places, not enough explanations are provided of what seem to be important aspects of the analysis (e.g., removal of data affected by great earthquakes is only mentioned in a suppl. Figure caption). As currently presented, I came away not being convinced yet that the proposed rate changes are significant. I am listing a number of concerns below, and various additional comments, questions and edit suggestions are in the annotated manuscript I am returning.*

190 *In the hope of a clarified version being more convincing, I recommend major revisions of the current manuscript, allowing for reworking and strengthening of the analysis. I would be glad to review a revised version and the authors could also contact me directly if I may have misunderstood or missed something important.*

195 We are deeply thankful to Roland for reviewing our manuscript and highlighting the points that needed to be clarified. We agree with him that we should have been more rigorous concerning the presentation of the data and the declustering method. We hope that we brought enough clarifications on the data used, the method and the Figures. Furthermore, we modified the abstract and introduction to mention the post-seismic Tohoku-oki consequences with more details and other observations of change in intermediate-depths earthquakes rates during the interseismic period. We also made a special effort about the declustering Figures in the Supp. Mat. section to clarify the data used in every Figures and removed two of them to improve clarity. We also added a Figure and two tables to complete the Coulomb stress change analysis.

205 *Itemized comments (unfortunately there were no line numbers in the manuscript file, but I hope the annotated manuscript from which these comments were derived will allow for clarifying these):*

210 Thank you for annotating the manuscript and having made the effort to itemize your questions/remarks. Sorry about the absence of line numbers, we added them in the manuscript.

p. 2, Abstract:

Not sure if this point about (still tentative) precursor suggestions is the best way to start off this paper.

215 *Why are you leaving out postseismic relaxation of coseismic stress changes in the mantle, which is the best documented process producing post-earthquake deformation transients?*

220 Thank you for pointing that out. Actually, we mention the possible pre-seismic phase because, if true, it might have affected a large-scale area, as is the case with our observations of the post-seismic phase on the deep seismicity. However, you are right to say that we are leaving out postseismic relaxation of coseismic stress changes in the mantle, so we added it in the abstract and

introduction. We also mention the few cases where interactions between shallow and deep parts of the subduction have been observed at different periods of the seismic cycle and updated the references with the recently published paper by Wimpenny, S., Craig, T., & Marcou, S. (2023).

225 Wimpenny, Sam, Tim Craig, and Savvas Marcou. "Re-Examining Temporal Variations in Intermediate-Depth Seismicity." *Journal of Geophysical Research: Solid Earth* 128.6 (2023): e2022JB026269.

230 *Use "background rate" here and/or elaborate to note that this is looking at "declustered seismicity".*

Modified in the main text l. 22 and 24

235 *p. 3 - top: As for the abstract, I am not sure that starting out talking about proposed precursory seismicity, deformation and gravity-field changes before large events is a good way to start out this paper.*

We elaborate on that in the introduction to emphasize the post-seismic observations of the Tohoku-oki earthquake and other interactions that might exist during the interseismic period.

240

As noted in the next sentence, the paper goes on to explore seismicity as a proxy of stress conditions after the Tohoku-oki earthquake, not before. Nonetheless, the cumulative event number plots presented in this paper don't seem to suggest notable rate changes in slab seismicity that could be related to the proposed months-long or multi-decadal precursors, right?

245

You are right, we don't seem to see a change of deep seismicity in the months or year preceding the Tohoku-oki earthquake at such large depths. We added a sentence in that sense l.158-159 in the main text.

250 *The studies noted here do not describe a (single) "preseismic phase", but range from investigations of decadal variations in plate interface coupling to (still debated) months-long slab deformation transients inferred from GRACE and GNSS time series.*

255 Yes, as we mentioned above, if these processes are true, their large-scale is interesting to us as we observe a large-scale consequences post-seismically to the Tohoku-oki earthquake on the deep seismicity.

260 *p. 4: Would it be possible to assess how sensitive the results are to the choice of declustering approach (using method of Marsan et al. , 2017) by comparing the results with those based on other approaches/algorithms (e.g., Zhuang et al. 2002, Zaliapin and Ben-Zion, 2015).*

The method by Marsan et al 2017 is very similar to the one of Zhuang et al. 2002, ie, it is based on a space-time ETAS model with a spatially varying background rate.

We run the Zaliapin et al (2006) method for:

- 265 - (1) all M2.5+ earthquakes from 1/1/2000 to 11/3/2013 (2 years after the M9), with no constraints on depth
- (2) all deep ($z > 150$ km) M2.5+ earthquakes from 1/1/2000 to 1/1/2021

270
275
280
285
290 *Figure 3: Computation of the declustering method by Zaliapin et al., 2006 for all M2.5+ earthquakes from 1/1/2000 to 11/3/2013 (blue line), all deep ($z > 150$ km) M2.5+ earthquakes (pink line).*

As the figure shows, in neither case is the distribution bimodal. Case (2) (deep earthquakes) is especially mono-modal, as there are very few aftershocks at these depths. We thus believe this method is not appropriate for declustering our dataset. Moreover, our results are little dependent on the declustering, as the conclusions remain the same even if we do not decluster (again, this is because there are very few aftershocks at those depths).

300 *Are there depth-dependent and lateral changes in M_c that could affect the results?*

In Figure 1 (extracted from Schorlemmer et al., 2018) we show that the M_c changes with depth but not so much laterally for a given depth. We are rather conservative taking a M_c of 3.5.

305 *"We detect no significant effect of the Tohoku-oki earthquake on the deep seismicity" Can you please specify what you mean by "deep seismicity" in this sentence; i.e., what depth range (> 150 km?).*

Actually, we here refer to the seismicity underneath the Japan Sea. We clarified this in the main text.

310 *p. 8: But isn't the deep seismicity not on the plate interface (or slab top, but in seismic planes of the double seismic zone within the plate with contrasting mechanisms and nodal planes very different*

from the slab top (e.g., Kita et al., 2010)? That is, it is not a matter of rake variations only. There is no "fault" at the top of the slab that these events relate to.

315 That is a very good point. In the manuscript, we compute the Coulomb stress change on two points that are on the top of the slabs because we make the simple hypothesis that if the stress increases, the velocity of the slab can increase and thus it would affect the forces inside the slab. We thus perform this analysis, to assess the sign of the Coulomb stress change that seems coherent with the strikes of the earthquakes in these areas. In reality, it is true that it a more complex and a better
320 model is needed.

However, you are correct to point out that the dips and strikes can greatly vary and we performed an extra analysis using different dips and strikes to see the effect on the Coulomb stress change values.

We selected the earthquakes for which we have the focal mechanisms that are closest to the two
325 points. For the point underneath Hokkaido, there are 12 earthquakes and 25 for the point at 34°N latitude (see Tables 1 and 2, Figure 4). Given the small number of focal mechanisms available at that depths, we keep earthquakes that occur both before and after the Tohoku-oki earthquake. We performed the same analysis than in the main text (ie 50m of slip on one fault) using the strike, dip and rakes of the earthquakes (Figure 4). Underneath Hokkaido, using the focal mechanisms of the
330 earthquake for the point geometry, we mainly find positive Coulomb stress changes but the results are disparate. Note that there is an earthquake with a strike much smaller than the other ones (point 6). Underneath Chiba, in the South, the Coulomb stress change is mainly negative, meaning an unloading of the point due to the 50m of slip that we use.

We added Figure 4 and the two tables in the Supplementary Material and a sentence to present them
335 in the main text l. 178-181.

Figure 4: a) Location of the earthquakes used to compute the Coulomb stress change color coded with depth. We find only 12 earthquakes close to the points in the North and 25 in the South. We show the 50m of slip dislocation used for the computation; b) Coulomb stress change of the earthquakes located underneath Hokkaido, we show their focal mechanisms color coded with depth. c) same for the south. This show that the Coulomb stress change is mostly positive for earthquakes underneath Hokkaido (ie promoting slip) except for three southernmost ones. We find mostly negative Coulomb stress change (ie preventing slip) with very diverse focal mechanisms. Two earthquakes that are the northernmost ones present positive Coulomb stress changes.

First time GPS analysis is mentioned? Shouldn't there be an introduction of the data?

340

We added it in the main text l.181-184.

When considering the postseismic GPS motions, it is not clear if the authors look for processes above and beyond the postseismic relaxation that model studies put forward to explain most of the deformation signal. Are co- and postseismic stress changes relevant?

345

The GPS signal that we observe is likely related to postseismic processes (afterslip + visco-elastic relaxation). However, in this study, we did not aim to model the post-seismic deformation but simply quantify the velocity field change after the Tohoku-oki earthquake. The changes in the seismicity rate occurs at the time of the megathrust earthquake which indicate that the co-seismic stress change might have played a role but there is also a long-term relaxation that we think is linked to the post-seismic visco-elastic relaxation.

350

355 *"viscoelastic relaxation of both oceanic and continental plates" : The viscoelastic relaxation occurs mostly in the mantle asthenosphere in the mantle wedge and below the oceanic plate (e.g., Freed et al., Hu et al., 2016), not in them as implied in this sentence. Good to clarify this.*

Thank you for pointing that out, we modified the text in that sense.

360 *"and could be facilitated by a low-viscosity channel," Is this envisioned low-viscosity channel on top of the slab? The coseismic stress changes that could drive such shear under Hokkaido must be quite small. It would be good to assess this scenario in a (simple) model. Some postseismic studies (e.g., Hu et al., 2016) did include a low-viscosity shear zone to represent such deep localized shearing. Unfortunately, this discussion of a low-viscosity channel and possible rheologies involved*
365 *comes across as highly speculative without actual modeling to test this idea.*

Several studies have invoked low viscosity channels both above and below the slab. In the analysis of Klein et al., 2017 in Chile the low-viscosity channel is located at the top of the slab and extends down to 150km. In Freed et al., 2017, the authors focus on the Honshu island only and locate a
370 weak lower slab beneath the outer rise domain. Hu et al., 2016, also focused on the Honshu island and propose a 2 km thick weak shear zone attached on top of the megathrust fault.

We agree that the coulomb stress change is small, and this precisely the reason why we think that a low viscosity channel could be a good candidate to explain our observations since we ‘need’ the slab to be transiently decoupled from the surrounding mantle to transfer the stress and the
375 deformation that far down along the slab.

Page 15:

380 *I am not a statistical seismologist, but this method seems sound. Based on my own experience, it is always useful to consider how the choice of declustering method affects the results. I think it would be worth running some comparisons.*

Can you briefly elaborate on what this comparison implies?

385 *P. 16: M 2.5 is well below Mc. How does using an incomplete catalog come into play?*

For declustering purposes, it is better to keep earthquakes smaller than M_c as potential “mainshocks”, as they affect the branching structure of mainshocks-aftershocks of ETAS models, see Sornette and Werner (2005). This simply adds extra information. We anyway only keep $m > 3.5$
390 declustered earthquakes eventually.

Sornette, D., & Werner, M. J. (2005). Apparent clustering and apparent background earthquakes biased by undetected seismicity. *Journal of Geophysical Research: Solid Earth*, 110(B9).

395 *Again, it would be good to evaluate spatio-temporal variations of M_c to avoid biases in the analysis.*

400 *"except the remarkable case of the 2011 Tohoku-oki mainshock itself": Wouldn't this "imperfect" declustering of the M9 aftershocks potentially affect the results of this study, possibly introducing a bias?*

Actually, since we look at the very deep seismicity, we are sure that the data are not polluted by the Tohoku-oki's aftershocks.

405 *"Figure S4 shows the selected deep earthquake dataset analyzed in this study, before and after declustering". Do you mean Figure S3? What is the "subset" mentioned here. In caption of Fig. S5 you "Note that we remove the area close to the Tohoku-oki and the Mw8.0 2003 Tokachi-oki rupture", but I think this is not explained otherwise in text. Is the gap in events at the 37-40 latitude range in Fig. S3 due to this? What does Fig. S3 look like applied to the original catalog of events?*

410 The gap at the 37-40 latitude range corresponds to the absence of deep seismicity underneath the Japan Sea. Please, note that we changed the figures concerning the declustering and their captions for more clarity.

415 *p. 20: "The change of velocity before and after the Tohoku-oki earthquake are simply calculated as the difference of slope in each time series, between a linear trend before the earthquake from 2008/01/01 to 2011/03/08 and a linear trend after the earthquake from 2011/03/13 to 2013/03/10." A linear trend and average velocity are not a very good representation of the rapidly decaying postseismic deformation in the first 2 years.*

420 As we mention above, we do not aim to model the postseismic phase but rather the change of velocity of the GPS data, using the same approach as for the seismicity rate analysis, ie a difference of the linear approximations after vs before the Tohoku-oki earthquake.

425 *Figure 2: The two example time series show similarly large changes in rate at times other than the M9, including a rate increase in 2006, a much larger decrease starting in ~2008 for 34N and sudden decrease in early 2010 for 44N. Do you consider these to be significant?*

430 You are correct to say that variations in seismicity rate can occur at any time. In this paper, we have endeavored to analyse in a statistical way how these changes can be important by computing 1) the probability to observe a seismicity rate change at the time of the Tohoku-oki earthquake, 2) the probability of such change in seismicity rate at a given time and, 3) the probability that the seismicity rate changes at any time. The Bayesian approach is very useful to compute the latest (see
435 Fig. 3). We see that the probability of having a change in seismicity rate fluctuates. For example, there is a large probability of a change for the data of latitude $< 28^\circ$ in 2008. The stack shows that the probability of having a change at the time of Tohoku-oki earthquake is large enough to be significant.

440 *Figure 4: Figures and their captions should be as self-explanatory as possible. Can you please elaborate in the caption on what this stress change is due to (coseismic elastic stress changes).*

We modified the figure caption.

445 *The assumption of receiver faults being on the slab surface and looking at varying rake angles alone is not warranted for intraplate deep-focus earthquakes.*

See our answer above.

450 *Fig. S1: Catalog is incomplete below $M_c = 3.5$. Could spatio-temporally variable M_c affect these plots and analysis.*

See our answer above.

455 *Good to include figures like Fig. S1 and S2 spanning full time period.*

We include a figure that shows the $m \geq 3.5$ earthquakes before / after declustering, for 2000 - 2015 (note that the declustering is exactly the same as in the paper, and is done using all $m \geq 2.5$ earthquakes). We only keep $m \geq 3.5$ events here, and not $m \geq 2.5$ because the figures are too busy at this scale for $m \geq 2.5$. It is clear that the smaller aftershock sequences are indeed well removed, but that the M9 aftershocks are not, as already commented in the MS.

460 *Are events in these two figures same as Fig. 4, but prior to removing events near 2003 and 2011 mainshocks?*

465 *Why not make more panels and show all the data available in this fashion.*

Good suggestion; as mentioned above, we have indeed added this figure in the suppl mat (Figure S1).

470 *These are also deep earthquakes?*

In Figure S1, these are all the JMA earthquakes, with no condition on depth. We have clarified this point in the caption.

475 *Point to evidence of M9 aftershocks not having been removed.*

Yes, but this is only true for shallow activity, not for deep earthquakes.

480 *What is the "subsetting" procedure to produce Fig. S3? Why not make a version of Fig. S3 with all data prior to this procedure, so we can better see the M9 aftershock effect?*

485 This figure (now Figure S2) only shows the deep earthquakes (depth > 150 km, $m \geq 3.5$), as we
now recall in the revised caption. There is no obvious M9 aftershock effect at this depth range; our
study is intended to reveal the small but significant changes in activity at these depths after the M9.

490 *"Fig. S3. Deep earthquakes selected in this study," : Please make clear if earthquakes in Figs. S1
and S2 not deep earthquakes. It seems that this would be the most important figure in the paper,
given that this shows all the data this study is based on, right?*

We now clearly state in the caption of Figure S1 that it displays all earthquakes, not conditioned on
their depth, for M3.5+.

495 *Why are there almost no earthquakes between ~37-40N, i.e., the latitude range of the M9
mainshock. Only after I get to the caption of Fig. S5 do I see mention of this representing a subset.*

As mentioned above, there is indeed a scarce seismicity underneath the Japan Sea.

500 *Fig. S5: Is this plotting the "subset" of data shown in Fig. S3? If so, why do we still see strong rate
increase W of the M9 rupture zone?
It is rather disconcerting that the rate increase in the N is invisible prior to declustering. What does
this figure look like without the removal of events near 2003 and 2011 mainshocks? Declustering
505 should make a lumpy quake distribution smoother, by removing aftershocks concentrated near their
mainshocks. That's not what we see here, and the entire pattern changes, with red turning blue and
blue turning red, while the overall smoothness is about the same or a little improved. I realize this is
likely related to the sampling scheme around the reference points.*

510 Thank you very much for raising that point because there was a mistake in the code that generated
the figure without declustering. We corrected it and now, the Figure S4 shows the same trend as the
declustered one and this is very reassuring since the intermediate and deep seismicity are known to
have less aftershocks as you mention. We see that the signal is actually stronger without
515 declustering but we think that it is important to decluster the data anyway not to be biased by the
few mainshock-aftershock sequences that can be present in the data.

*This is not described in the text. Does this procedure explain the gap in events in Fig, S3? What are
the exact areas that got removed.*

520 See our answer above and the main text.

Why also remove the Tokachi earthquake area, given how early this event was?

It also greatly affected the shallow seismicity. We rephrase the text to clarify that, in this study, we
actually look at the earthquakes from 150km to 680km and do not analyse the shallow seismicity at
525 all, since it is too affected by the Tokachi and the Tohoku-oki earthquakes, even after declustering.

*Figure S6: Can you add a 300-km circle in lower right to provide a sense of scale for these
sampling area.*

530 We added a circle but, please note that the dip of the slab varies greatly and that the circle of 300km
of distance around a point might appear distorted.

What is the actual number of events within each of these latitude windows. What do cumulative

535 *number plots look like if rather than stacking the greatly overlapping samples around the reference points, you simply provide the count for each latitude interval? That seems like a more straightforward way of displaying the results.*

Thank you Roland, we did the analysis you suggested and added it in the Supplementary Material and splitted the Figures for more clarity. We did the same for the declustered 4.5 magnitude earthquakes and the USGS catalogue.

540 *Just looking at deviation from straight line (drawn in), rates seem to start to increase in ~2009, with some variability.*

545 See our answer above concerning the Bayesian pdf estimation.

Fig. S12/S13: Is this analysis of $M > 4.5$ events also on declustered events and after removing events near 2003 and 2011 mainshocks? As I suggested there, can you repeat this analysis by simply plotting cumulative number of all events in the three latitude ranges shown in Fig. S6.

550 *Fig. S12 & S13: Even though they are supposed to both show $M > 4.5$ events and include 70 events, there are about twice as many events in the NEIC catalog plots than the JMA catalog shown here. Why is that?*

555 There was a mistake in the Figure caption. We took the 50 closest earthquakes for the M4.5 JMA figure and the 100 closest earthquakes for the USGS figure. Now, we changed the Figure to show the three latitude ranges as shown in Fig. S12 to S13, as you suggested so we take all the earthquakes with magnitude ≥ 4.5 in the three different latitude ranges.

560 *Fig. S13: Both areas seem to show a rather pronounced rate decrease in the (~5) years before Tohoku-oki.*

Roland Burgmann

2010	8	24	13	50	0	44.3	141.2	230	156	36	-50	291	63	-115	0.058	0.03	0.07
2013	1	22	13	29	0	44.5	141.1	280	188	58	-30	295	65	-144	-0.014	-0.046	-0.032
2016	1	11	17	8	1	44.4	141.2	265	178	51	-27	286	69	-138	0.021	-0.031	0.009
2017	1	15	12	13	21	44.3	141.2	251	85	65	151	188	64	28	-0.004	0.237	0.091
2005	9	20	11	59	0	43.6	140.5	195	103	66	177	194	88	24	-1.816	2.862	-0.671
2005	12	11	8	25	0	44.8	141.0	280	75	18	-89	254	72	-90	0.973	0.816	1.299
2009	7	10	18	16	0	44.5	141.5	250	247	10	160	357	87	80	0.038	0.06	0.062
2010	7	19	8	21	0	44.6	141.3	290	176	57	-19	276	75	-145	0.347	-0.328	0.216
2016	8	5	3	30	45	43.3	140.2	195	123	36	179	213	89	54	-2.14	1.329	-1.60943
2017	1	7	9	35	14	43.8	140.7	222	169	34	-96	356	56	-86	-0.605	-0.02	-0.614
2017	9	9	10	12	44	44.5	141.0	276	183	36	-34	301	71	-121	0.136	-0.117	0.089
2018	10	25	10	25	35	44.8	141.3	279	175	46	-8	270	85	-135	0.528	-0.136	0.474

565 Table 1 : Location and focal mechanisms of the 12 earthquakes closest to the point underneath Hokkaido. Format : year, month, day, hour, minute, second, latitude, longitude, depth (km), strike 1, dip 1, rake 1, strike 2, dip 2, rake 2, shear stress (bar), normal stress (bar), Coulomb stress (bar).

570 2002 5 15 10 23 0 34.6 139.4 140 71 35 151 185 74 58 0.134 -0.114 0.088

2003	10	19	5	21	0	34.6	137.9	260	82	88	-99	338	9	-14	0.121	-0.047	0.102
2004	3	20	16	18	0	34.1	139.4	165	19	84	1	289	89	174	0.101	-0.417	-0.066
2004	9	4	6	12	0	34.1	138.3	280	37	30	95	211	60	87	0.135	0.7	0.137
2004	11	3	20	13	0	33.8	138.7	260	93	48	154	201	71	46	0.057	-0.038	0.041
2005	5	28	20	55	0	34.5	137.7	320	40	88	-150	308	60	-2	-0.011	-0.163	-0.077
2006	3	4	18	38	0	33.8	137.6	340	201	82	106	317	17	27	0.016	-0.138	-0.040
2006	4	28	22	16	0	34.4	138.1	320	86	46	-111	294	48	-70	-0.037	-0.047	-0.056
2007	1	15	18	18	0	35	138.9	180	93	39	-169	354	83	-52	0.021	-0.082	-0.012
2007	7	28	8	55	0	34	137.6	340	191	80	104	316	17	37	-0.011	-0.125	-0.061
2008	2	22	8	24	0	34.5	138	280	52	81	-131	311	41	-14	0.013	-0.18	-0.059
2008	3	5	16	2	0	34.2	137.9	320	228	84	98	353	10	36	0.065	-0.148	0.5
2009	6	17	16	24	0	33.90	137.5	340	53	21	142	179	78	73	0.014	0.022	0.023
2010	8	29	4	27	0	34.60	138.1	280	55	87	-104	311	14	-140	0.035	-0.186	-0.040
2011	1	22	2	48	0	34.50	137.9	290	51	90	-120	321	30	0	0.048	-0.173	-0.021
2013	9	23	8	7	0	33.80	137.5	380	87	58	-148	338	63	-37	0.044	-0.033	0.03
2014	5	4	20	18	0	35	139.4	160	208	75	-80	354	18	-123	0.043	-0.484	-0.15
2015	3	13	15	12	48	34.7	138.5	218	85	30	170	184	85	60	-0.7	-0.018	-0.014
2016	1	18	0	56	48	35.2	139.27	150	72	71	50	321	43	152	-0.057	-0.372	-0.206
2017	7	18	7	47	1	34.5	137.6	316	205	70	127	319	41	31	0.010	-0.136	-0.045
2019	4	29	5	24	0	34.5	137.6	318	226	74	155	323	66	18	0.082	-0.124	0.028
2020	2	2	6	28	8	35.5	138.9	180	218	88	-49	310	41	-178	0.449	-0.212	0.364
2020	6	21	4	55	19	33.9	138.0	317	60	57	-167	323	79	-34	0.041	-0.102	0.1
2020	7	24	19	51	33	35.7	138.8	162	46	51	37	291	62	135	0.709	0.242	0.805
2020	10	6	23	2	6	34.4	138.0	302	242	83	144	337	54	9	0.164	-0.110	0.120

Table 2: Location and focal mechanisms of the 12 earthquakes closest to the point underneath 34°N latitude. Format : year, month, day, hour, minute, second, latitude, longitude, depth (km), strike 1, dip 1, rake 1, strike 2, dip 2, rake 2, shear stress (bar), normal stress (bar), Coulomb stress (bar).

REVIEWERS' COMMENTS

Reviewer #2 (Remarks to the Author):

Gardonio et al. have done a thorough job addressing the comments by both reviewers, clearly documented in the rebuttal letter and the annotated revised manuscript. I only have a few remaining comments and suggestions listed below, amounting to minor revisions.

Line 24: to 2021/3/11? Seismicity until March 11, 2021 is considered in Fig. 1 and related text. It would be good to have the cumulative time series plots shown in the paper also extend through that full time period. This will allow readers to make out the temporal evolution described in reference to that figure in lines 94-97.

Line 26: It is not clear what is meant by "coherent" here. Do you mean consistent with the spatial extent of the resolved postseismic deformation transients inferred from the GPS observations? Can you please clarify the wording.

Line 37-41, 50-52: I am still a bit puzzled by the emphasis put in the introduction on still-debated precursory slab deformation processes, when this paper is about how "large subduction earthquakes can affect the plate stress regime". In particular, I wonder about highlighting the precursory regional "wobbles" in GNSS time series put forward by Bedford et al. (The observation of large-scale surface displacements reversals, several months before megathrust earthquakes might also suggests the existence of deep precursors¹¹.), given a coauthor of this study (Anne Socquet) found that the proposed time-varying signals in the pre-Tohoku period are processing artifacts. The authors might want to discuss the best wording of this paragraph.

Line 43: Add "postseismic" for "a postseismic increase" to make clear that now you are considering post-earthquake changes.

Line 58 and 204: Is ref. 38 the wrong reference here (about Chile)? Did you mean to cite post-Tohoku deformation studies, such as 34, 35?

Line 148: minimum magnitude (M_c remains 3.5)

Line 173-180: I still think that computing stress on the deep plate interface and reference to "promoting slip" is not really a good way to do this. The addition of example stress calculations using the focal mechanisms is good.

Line 180-182/Fig. S14: While shear-stress changes are identical on the two orthogonal planes, normal stresses generally are not. Did you compute stress on both nodal planes or pick one or the other based on some rationale?

Fig. 1:

- I agree with Reviewer 1 that given the highly non-linear nature of postseismic deformation (and seismicity) comparing cumulative GPS displacements during pre- and post-earthquake periods of equal duration in Fig. 1A would be preferable. Other readers would likely be concerned about this as well.
- The log-scaling of the GPS displacement vectors is not helpful. It is almost impossible to make out what the amplitudes of the postseismic displacements are in this depiction. If you want to highlight the modest but important motions at large distances, using two different scales for near- and far-field motions and displaying those in two different colors, would probably work better.
- Figs. 1b&c show results extending through longer time periods (through 03/11/2017 and 03/11/2021), however, these are not shown in the time series in later figures. Could you please update these figures to better illustrate the temporal evolution of the inferred rate changes mentioned in lines 94-97. Does the acceleration pattern still appear when making a map for 2013/03/11 to 2017/03/11 and 2017/03/11 to 2021/03/11?
- I am curious why later figures only show results until 2013/03/11, especially the time series in Fig. 2b, 3, S6, S12, S13.

Figure 4: I continue to be concerned about the assumption of a common attitude (allowing for variable rake) of receiver faults in the CFF calculations.

Fig. S1: Point to Fig. S2 at end of caption (... 150 km (see Fig. S2).). Why does Fig. S1 include data until 2015, Fig. S2 only to 2013?

Fig. S6, S12, S13: Please show time series through March 2021? Can you also add a dashed line with slope following linear fit to pre-2011 data, as in Fig. 2.

Figure S14: Why is there a gradient in gray shading of the rupture rectangle? That might make some people think a distributed slip model was used.

Roland Burgmann

REVIEWERS' COMMENTS

Reviewer #2 (Remarks to the Author):

Gardonio et al. have done a thorough job addressing the comments by both reviewers, clearly documented in the rebuttal letter and the annotated revised manuscript. I only have a few remaining comments and suggestions listed below, amounting to minor revisions.

Line 24: to 2021/3/11? Seismicity until March 11, 2021 is considered in Fig. 1 and related text. It would be good to have the cumulative time series plots shown in the paper also extend through that full time period. This will allow readers to make out the temporal evolution described in reference to that figure in lines 94-97.

In the paper, we mainly focus on the two years period that follows the Tohoku earthquake because the effect is the strongest at that time and it is interesting to compute the probability of seismicity rate change for that period. However, we agree that it is important to show how the seismicity rate evolves with time, thus, we added a Figure in the Supplementary Material (Fig. S7) to show the cumulative number of events for the seismicity of latitude 44°N (in red) and 34°N (in blue) up to 2021/3/11.

Line 26: It is not clear what is meant by "coherent" here. Do you mean consistent with the spatial extent of the resolved postseismic deformation transients inferred from the GPS observations? Can you please clarify the wording.

We changed coherent for consistent.

Line 37-41, 50-52: I am still a bit puzzled by the emphasis put in the introduction on still-debated precursory slab deformation processes, when this paper is about how "large subduction earthquakes can affect the plate stress regime". In particular, I wonder about highlighting the precursory regional "wobbles" in GNSS time series put forward by Bedford et al. (The observation of large-scale surface displacements reversals, several months before megathrust earthquakes might also suggests the existence of deep precursors¹¹.), given a coauthor of this study (Anne Socquet) found that the proposed time-varying signals in the pre-Tohoku period are processing artifacts. The authors might want to discuss the best wording of this paragraph.

We added a sentence l.41 : « However, this is highly debated in the community since it might be due to processing artifacts.»

Line 43: Add "postseismic" for "a postseismic increase" to make clear that now you are considering post-earthquake changes.

Thank you for, we added it.

Line 58 and 204: Is ref. 38 the wrong reference here (about Chile)? Did you mean to cite post-Tohoku deformation studies, such as 34, 35?

Yes, it is a reference to Chile. We added « as suggested in other subduction zone» at l. 58 to clarify our point.

Line 148: minimum magnitude (Mc remains 3.5)

Yes, we changed the text.

Line 173-180: I still think that computing stress on the deep plate interface and reference to "promoting slip" is not really a good way to do this. The addition of example stress calculations using the focal mechanisms is good.

We agree with this point but computing the stress change is something expected when considering the effect of an earthquake on the surrounding seismicity. However, here the changes are very small and the stress change computation is here to give first order results. We change « promoting slip » in « would promote slip” l. 178.

Line 180-182/Fig. S14: While shear-stress changes are identical on the two orthogonal planes, normal stresses generally are not. Did you compute stress on both nodal planes or pick one or the other based on some rationale?

Thank you for pointing that out. Initially, we took the first nodal plane but we computed for the second and added it in Figure S14 and specify it in the main text.

Fig. 1:

- I agree with Reviewer 1 that given the highly non-linear nature of postseismic deformation (and seismicity) comparing cumulative GPS displacements during pre- and post-earthquake periods of equal duration in Fig. 1A would be preferable. Other readers would likely be concerned about this as well.

Actually the time periods used in this study are comparable (2008/1/1 to 2011/3/11 vs 2011/3/11 to 2013/3/11). The cumulative GPS displacement is stable in the time period before the Tohoku earthquake so taking a shorter duration of 2 years instead of 3 will not change the results.

- The log-scaling of the GPS displacement vectors is not helpful. It is almost impossible to make out what the amplitudes of the postseismic displacements are in this depiction. If you want to highlight the modest but important motions at large distances, using two different scales for near- and far-field motions and displaying those in two different colors, would probably work better.

Thank you for this solution, we changed the figure accordingly.

- Figs. 1b&c show results extending through longer time periods (through 03/11/2017 and 03/11/2021), however, these are not shown in the time series in later figures. Could you please update these figures to better illustrate the temporal evolution of the inferred rate changes mentioned in lines 94-97. Does the acceleration pattern still appear when making a map for 2013/03/11 to 2017/03/11 and 2017/03/11 to 2021/03/11?

We added a figure in the supplementary material to show the time series of the earthquakes around 44°N latitude and 34°N latitude up to 2021/3/11 (Fig. S6).

- I am curious why later figures only show results until 2013/03/11, especially the time series in Fig. 2b, 3, S6, S12, S13.

As already mentioned above, the effect of the Tohoku-oki earthquake is the strongest during this time period thus we focused the statistical analysis on this time period as said in l. 101-102 in the main text.

Figure 4: I continue to be concerned about the assumption of a common attitude (allowing for variable rake) of receiver faults in the CFF calculations.

See our answer above.

Fig. S1: Point to Fig. S2 at end of caption (... 150 km (see Fig. S2).). Why does Fig. S1 include data until 2015, Fig. S2 only to 2013?

As already mentioned above, we focus on the 2011/3/11-2013/3/11 period and Fig. S2 corresponds to all the earthquakes used in the statistical analysis. We changed Fig. S1 to show all the earthquakes up to 2021/3/11.

Fig. S6, S12, S13: Please show time series through March 2021? Can you also add a dashed line with slope following linear fit to pre-2011 data, as in Fig. 2.

We changed the figure accordingly.

Figure S14: Why is there a gradient in gray shading of the rupture rectangle? That might make some people think a distributed slip model was used.

Thanks for pointing that out, we removed it.

Roland Burgmann